# Diagnostic Utility of Selected Matrix Metalloproteinases (MMP-2, MMP-3, MMP-11, MMP-26), HE4, CA125 and ROMA Algorithm in Diagnosis of Ovarian Cancer

**DOI:** 10.3390/ijms25116265

**Published:** 2024-06-06

**Authors:** Aleksandra Kicman, Ewa Gacuta, Monika Kulesza, Ewa Grażyna Będkowska, Rafał Marecki, Ewa Klank-Sokołowska, Paweł Knapp, Marek Niczyporuk, Sławomir Ławicki

**Affiliations:** 1Department of Aesthetic Medicine, The Faculty of Pharmacy, Medical University of Białystok, 15-267 Białystok, Poland; olakicman@gmail.com (A.K.); niczy.ma@gmail.com (M.N.); 2Department of Perinatology, University Clinical Hospital of Bialystok, 15-276 Białystok, Poland; sunnyeve@wp.pl; 3Department of Population Medicine and Lifestyle Diseases Prevention, The Faculty of Medicine, Medical University of Białystok, 15-269 Białystok, Poland; monika.kulesza@sd.umb.edu.pl; 4Department of Haematological Diagnostics, The Faculty of Medicine, Medical University of Białystok, 15-269 Białystok, Poland; grazyna.bedkowska@umb.edu.pl; 5Department of Psychiatry, The Faculty of Medicine, Medical University of Białystok, 15-272 Białystok, Poland; marecki.rafal.96@gmail.com; 6University Cancer Center, University Clinical Hospital of Bialystok, 15-276 Białystok, Poland; ewaklank@wp.pl (E.K.-S.); pawel.knapp@umb.edu.pl (P.K.)

**Keywords:** MMP-2, MMP-3, MMP-11, MMP-26, HE4, CA125, ROMA algorithm, ovarian cancer, *Serous cystadenomas*, plasma concentration

## Abstract

Ovarian cancer (OC) has an unfavorable prognosis. Due to the lack of effective screening tests, new diagnostic methods are being sought to detect OC earlier. The aim of this study was to evaluate the concentration and diagnostic utility of selected matrix metalloproteinases (MMPs) as OC markers in comparison with HE4, CA125 and the ROMA algorithm. The study group consisted of 120 patients with OC; the comparison group consisted of 70 patients with benign lesions and 50 healthy women. MMPs were determined via the ELISA method, HE4 and CA125 by CMIA. Patients with OC had elevated levels of MMP-3 and MMP-11, similar to HE4, CA125 and ROMA values. The highest SE, SP, NPV and PPV values were found for MMP-26, CA125 and ROMA in OC patients. Performing combined analyses of ROMA with selected MMPs increased the values of diagnostic parameters. The topmost diagnostic power of the test was obtained for MMP-26, CA125, HE4 and ROMA and performing combined analyses of MMPs and ROMA enhanced the diagnostic power of the test. The obtained results indicate that the tested MMPs do not show potential as stand-alone OC biomarkers, but can be considered as additional tests to raise the diagnostic utility of the ROMA algorithm.

## 1. Introduction

According to GLOBOCAN 2020, ovarian cancer (OC) is the seventh most common cancer in the female population (3.4% of all cancers in women) and the eighth leading cause of death in women worldwide. At the same time, OC is one of the most common gynecological cancers—it ranks third in terms of incidence, just behind cervical and endometrial cancer [1,2,3]. OC is a heterogeneous disease, and based on histopathology, it is divided into high-grade and low-grade serous, endometrioid, clear cell and mucinous carcinomas. Patients are most often diagnosed with OC of the high-grade serous carcinoma, which has the most unfavorable prognosis and accounts for most deaths in women with OC (70–80%) [2,4].

OC is often referred to as the “silent killer”, this is due to its mostly asymptomatic course, with some patients reporting non-specific, non-reproductive symptoms such as pelvic pain or discomfort, early satiety or frequent urinary urgency [2,5]. Because the ovaries are not surrounded by anatomical structures, tumor cells easily exfoliate and enter distant organs through the peritoneal fluid, forming metastatic foci [2,6]. The ease of metastasis formation and asymptomatic course results in 60% of patients being diagnosed at a late stage of the disease, when metastatic foci occur. According to the International Federation of Gynecology and Obstetrics (FIGO) classification at stage III or IV [2,4,7,8]. All these factors translate into a high mortality rate for women with OC, with an estimated 5-year survival rate for patients in stage III of 27%, while in stage IV it is only 13% [2,7]. Additionally, the overall 5-year survival for women with OC ranges from 30% to 40% [2,9]. Detection of OC at an earlier stage translates into faster implementation of treatment and a better prognosis for patients. For disease detected at stage I, the 5-year survival rate is 90%, while at stage II it is 70% [1,9,10].

The unfavorable prognosis of patients is also compounded by the lack of effective screening tests. Currently, one of the few available diagnostic tests for OC consists of transvaginal ultrasonography (TVS) in combination with the determination of tumor biomarkers CA125 and/or HE4 [9,11,12]. Additionally, in patients with pelvic masses, a ROMA (Risk of Ovarian Malignancy Algorithm) is calculated which takes into account the patient’s menopausal status [10,13]. It is unfortunate that these tests have too low sensitivity and specificity to allow effective diagnosis of the disease [9,11,12]. The titers of both CA125 and HE4 concentrations are influenced by a number of physiological phenomena and pathological conditions, particularly those related to the reproductive tract [2,8,12].

Treatment of OC depends on its stage and on the basis of the histopathological diagnosis. Treatment usually includes surgery and the use of chemotherapy. Radiation therapy is most often not used to treat OC. Surgical treatment is aimed at confirming the diagnosis of OC, determining the stage of the disease and maximizing the removal of tumor lesions [14]. Chemotherapy treatment includes first-line therapy with carboplatin with paclitaxel; other agents include doxorubicin, topotecan or gemcitabine. For patients with BRCA1/BRCA2 mutations or recurrent OC, the PARP inhibitor Olaparib is also used. Response to treatment depends on the degree of OC, histologic type and sensitivity to platinum derivatives [1,4,14,15].

In 2022, OC was diagnosed in 325,000 women worldwide and, according to the International Agency for Research on Cancer, this number will continue to rise. In 2045, an estimated 477,000 women will develop OC, while 328,000 will die [1,16]. Given the ever-increasing number of OC cases and deaths, it is expedient to seek new tests that would detect OC at an earlier stage, thereby improving the prognosis of patients—for example, through the use of neoadjuvant chemotherapy [17]. Such postulated methods include the determination of tumor markers from peripheral blood [18,19,20,21].

Currently, there are more than a dozen compounds that are postulated as potential markers; however, the outstanding potential is characterized by MMPs [2]. MMPs belong to a group of proteolytic enzymes whose prolonged activity is associated with the process of carcinogenesis. In the initial stages of the process, they are responsible for the induction of genomic instability, through the induction of angiogenesis and the promotion of cancer cell proliferation and migration; MMPs contribute to tumor growth, while their proteolytic properties are important in the formation of metastasis [2,22,23]. In the course of OC, overexpression of various MMPs is observed, additionally, there are preliminary studies on the possibility of their use as markers of this disease [2,24,25].

The purpose of the present study was to determine the diagnostic utility of selected MMPs, MMP-2, MMP-3, MMP-11 and MMP-26, as new biomarkers in patients with ovarian cancer and to compare the results obtained with routinely determined markers CA125 and HE4 and the ROMA algorithm. The studied parameters were also determined in women with benign ovarian lesions (*Serous cystadenomas*) and in healthy women. This work is a continuation of our research team’s studies on the potential of MMPs as modern markers in ovarian cancer diagnosis [20,21].

## 2. Results

### 2.1. Plasma Concentrations of MMP-2, MMP-3, MMP-11, MMP-26 and Comparative Markers HE4 and CA125

Plasma concentrations of MMP-2, MMP-3, MMP-11, MMP-26, CA125 and the value of the ROMA algorithm expressed in percentages in ovarian cancer patients (OC) begin lesion patients (BL) and healthy subject (HS) are summarized in Table 1 and Figure 1, Figure 2, Figure 3, Figure 4, Figure 5 and Figure 6.

#### 2.1.1. MMP-2 Concentrations in the Study Groups

Higher MMP-2 concentrations were observed in the benign lesion (BL) group (median: 211.50 ng/mL) compared to the ovarian cancer (OC) group (median: 203.00 ng/mL, *p* = 0.12517), but this was not a statistically significant difference. Higher MMP-2 concentrations were observed in the BL group compared to the healthy subject (HS) group (200.195, *p* = 0.007795). This is shown in Figure 1.

#### 2.1.2. MMP-3 Concentrations in the Study Groups

MMP-3 concentrations in the OC (median: 9.32 ng/mL) and BL (median: 9.84 ng/mL) groups were comparable. At the same time, lower concentrations of MMP-3 (median: 7.97 ng/mL) were observed in the HS group compared to BL (*p* = 0.0001) and OC (*p* = 0.0003) (Figure 2).

#### 2.1.3. MMP-11 Concentrations in the Study Groups

The highest MMP-11 concentrations were observed in the OC group (median: 1.50 ng/mL) compared to BL group (median: 0.24 ng/mL, *p* < 0.000001) and HS group (median: 0.51 ng/mL, *p* < 0.000001); this is shown in Figure 3.

#### 2.1.4. MMP-26 Concentrations in the Study Groups

The highest MMP-26 concentrations were observed in the BL group (median: 10.44 ng/mL) compared to the OC group (median: 9.33 ng/mL, *p* = 0.5116) and HS group (median: 7.1625 ng/mL, *p* = 0.000009). The difference between MMP-26 concentrations in OC and BL groups were not statistically significant. The results are shown in Figure 4.

#### 2.1.5. Concentrations of the Comparative Markers HE4 and CA125 in the Study Groups

The highest concentrations of HE4 and CA125 were found in the OC group (HE4 median: 132.95 U/mL, CA125 median: 313.55 U/mL) compared to the BL group (HE4 median: 52.50, *p* < 0.000001; CA125 median: 11.253 U/mL, *p* < 0.000001) and HS group (HE4 median: 39.505 U/mL, *p* < 0.000001; CA125 median: 16.9 U/mL, *p* < 0.000001). This is shown in Figure 5 (HE4) and Figure 6 (CA125).

#### 2.1.6. ROMA Algorithm Values in the Study Groups

The highest ROMA algorithm values were shown in the OC group (median: 78.515%) compared to the BL group (median: 11.25%; *p* < 0.000001). These results are shown in Figure 7. All OC patients who were premenopausal were found to have a high risk of ovarian cancer, for OC patients after menopause, a high risk was calculated for 82 women (87.23% of patients), while a low risk was calculated for 12 women (12.77% of patients). In the BL group, for premenopausal patients, low risk was calculated for 28 women (90.32% of patients), while high risk was calculated for 3 patients (9.68% of patients). In BL patients after menopause, most women also had a low risk of OC diagnosis, with 35 patients (89.74% of patients), while a high risk was found for 4 patients (10.26% of patients). In a group of healthy women, all women were postmenopausal and had a low risk of OC (100% of patients).

### 2.2. Evaluation of Correlation by Spearman’s Method

Correlation evaluation of studied MMPs and comparative markers (HE4 and CA125) are shown in Table 2. The correlation analysis between the studied MMPs and the ROMA algorithm is shown in Table 3. Spearman’s non-parametric test was selected for analysis.

We found a statistically significant correlation between MMP-11 and HE4 in the OC group (r = 0.2220; *p* = 0.0148) and also a statistically significant negative correlation between MMP-2 and MMP-11 (r = −0.3445; *p* = 0.0143 in the HL group). Apart from the described correlations, no other statistically significant correlations were found between the studied parameters. When we performed correlation analysis between selected MMPs and the ROMA algorithm, the only statistically significant correlation we found was between MMP-11 and the ROMA algorithm in OC patients (r = 0.2026; *p* = 0.0265).

### 2.3. Diagnostic Criteria of MMP-2, MMP-3, MMP-11, MMP-26, HE4, CA125 and ROMA

The diagnostic criteria are as follows: diagnostic sensitivity (SE), diagnostic specificity (SP), positive predictive value (PPV) and negative predictive value (NPV) of all tested parameters for OC; patients are presented in Table 4.

The highest SE values of all tested parameters were obtained for CA125 (94.17%), while the highest value of this parameter among tested MMPs was characterized by MMP-26 (78.33%). None of the tested MMPs (MMP-2—42.5%; MMP-3—64.17%; MMP-11—70.83%) showed higher SE values than CA125. The SE values obtained for the tested MMPs were mostly lower than the SE for HE4 (76.67%); only for MMP-26 were higher SE values obtained than for HE4. The ROMA parameter showed a very high SE (90%), while surpassing the values of this parameter for all tested MMPs and HE4. Next, combined analyses were performed for the ROMA algorithm and the tested MMPs. Performing a combined analysis of the ROMA parameter with individual MMPs resulted in an increase in SE values, with the highest increase in SE found in the combined analysis of MMP-26 + ROMA (94.17%). Subsequently, analyses of the ROMA algorithm with two or three MMPs were performed. Performing the analysis of the ROMA algorithm with two MMPs was associated with a slight increase in SE (Table 5). The highest SE values were obtained for the combination of MMP-3 + MMP-26 + ROMA (95.83%). Also, performing ROMA algorithm analysis with three MMPs increased SE values, and the highest values of this parameter were found for the MMP-2 + MMP-11 + MMP-26 + ROMA analysis (95.83%). A single analysis of the combination of all the MMPs tested and the ROMA algorithm also increased the SE value (95.83%).

In the case of SP, the highest values of this parameter are found for CA125 (98%); slightly lower SP values were obtained for HE4 (92%). All tested MMPs had lower SP values (MMP-2—68%; MMP-3—68%; MMP-11—76; MMP-26—68%). SP values for the ROMA algorithm were identical to the SP for CA125 (98%). After performing a combined analysis of the ROMA algorithm with the two MMPs, the SP value increased only for the MMP-3 + ROMA analysis (100%). Performing the analysis of the ROMA algorithm with two MMPs was not associated with an increase or decrease in SP (Table 2), and the highest values of this parameter are found for the combination of MMP-2 + MMP-26 + ROMA and MMP-11 + MMP-26 + ROMA (96%). Performing a combined analysis of the ROMA algorithm with the three MMPs increased the SP value only for the MMP-2 + MMP-3 + MMP-26 + ROMA combination (100%). Combined analysis of all parameters tested was associated with a decrease in SP (96%).

For PPV, the highest values were obtained for CA125 (99.12%), slightly lower PPVs are found for HE4 (95.83%). The highest PPV among the MMPs studied is found for MMP-11 (87.63%). The PPV for all MMPs tested was lower than for the comparative markers and the ROMA algorithm (MMP-2—76.12%; MMP-3—82.80%; MMP-26—85.45%). The ROMA algorithm acquired high PPV values of 99.08%. Performing the analysis of the ROMA algorithm with individual MMPs was associated with a slight decrease or increase in PPV (Table 5); however, the combined analysis of the ROMA algorithm with MMP-3 was associated with an increase in this parameter up to 100%. Performing a combined analysis of the ROMA algorithm with the two MMPs was not associated with an increase in PPV. Combined analysis of the ROMA algorithm with three MMPs induced a slight increase in PPV when combining MMP-2 + MMP-3 + MMP-26 + ROMA (100%) and MMP-2 + MMP-11 + MMP-26 + ROMA (98.29%). The combined analysis of all MMPs with the ROMA algorithm translated into a slight decrease in PPV (98.29%).

The highest NPV values are found for the comparative markers CA125 (87.50) and HE4 (62.16%). All tested MMPs had lower NPV values compared to comparative markers (MMP-2—33.01%; MMP-3—44.16%; MMP-11—52.05; MMP-26—56.67%). The NPV determined for the ROMA algorithm (80.33%) was higher than all the MMPs and HE4 tested. Performing the analysis of the ROMA algorithm with single MMPs was associated with a slight increase in NPV (Table 5), with the highest value of this parameter obtained for the MMP-26 + ROMA analysis (87.27%). Analysis of the ROMA algorithm with two or three MMPs was also associated with an increase in NPV, with the highest value found for the combination of MMP-3 + MMP-26 + ROMA (90.57%) and MMP-2 + MMP-11 + MMP-26 + ROMA (90.57%). Combined analysis of all parameters was again associated with an increase in NPV (90.57%).

### 2.4. Evaluation of the Diagnostic Power of Tests (ROC Function)

By evaluating the area under the ROC curve (AUC), the diagnostic power of the tests was assessed. Performing an ROC analysis makes it possible to determine whether the chosen test distinguishes between normal and abnormal results. A diagnostically ideal test allows the complete differentiation of a healthy person from a diseased person, with a sensitivity value of 100% and specificity of 100% where the line in the ROC function will completely coincide with the *Y*-axis, and the AUC will reach a value of 1. A diagnostically useless test (which does not allow the differentiation of diseased from healthy people) will have a straight line inclined at an angle of 45 degrees to the *X*-axis and the AUC value will be close to 0.5—the limit of the diagnostic usefulness of the test. Characteristics of the ROC curve for tested parameters individually and in combination in OC patients are presented in Table 5.

All tested parameters, together with multivariate parameters, showed significant diagnostic power in relation to AUC = 0.5. The highest AUC values in OC patients were obtained for routine marker CA125 (AUC = 0.9918; *p* < 0.000001); similar values were obtained for HE4 (AUC = 0.9429; *p* < 0.000001) and the ROMA algorithm (AUC—0.9336; *p* < 0.000001). None of the tested MMPs exceeded the diagnostic power values of the comparative markers and the ROMA algorithm (Table 5). The highest AUC value among the tested MMPs is found for MMP-26 (AUC = 0.7751; *p* = 0.0003).

Performing a combined analysis of the ROMA algorithm with individual MMPs slightly increased the AUC compared to the diagnostic power of the ROMA algorithm (Table 6) in all cases except for the combined analysis of MMP-11 + ROMA (AUC = 0.9266; *p* < 0.000001). The highest diagnostic power of the test is observed for the MMP-2 + ROMA analysis (AUC = 0.9534; *p* < 0.000001). Also, performing a combined analysis of the ROMA algorithm with two MMPs increased the diagnostic power of the assay compared to that of the ROMA algorithm. The highest AUC was obtained for the combination of MMP-2 + MMP-26 + ROMA (AUC = 0.9580; *p* < 0.000001). Performing ROMA analysis with three MMPs or with all MMPs tested similarly to previous analyses increases the diagnostic power of the assay, and the highest AUC was obtained for the combination of MMP-2 + MMP-11 + MMP-26 + ROMA (AUC = 0.9650; *p* < 0.000001). When all parameters were analyzed simultaneously, the AUC was 0.9674 (*p* < 0.000001). AUC values obtained for all parameters and combinations between parameters are shown in Figure 8.

## 3. Discussion

Ovarian cancer (OC), has the most unfavorable prognosis among all gynecological cancers. This is mainly due to the high heterogeneity of the disease and the asymptomatic or scanty course of the condition. In addition, due to the lack of anatomical barriers around the ovary, tumor cells are distributed through the peritoneal fluid and easily form metastatic foci [1,2]. As a result, most patients are diagnosed at advanced stages of OC, III or IV, according to the FIGO classification. Patients in these stages have low survival rates [2,4,6,7]. Importantly, unlike cervical cancer (smear test) or breast cancer (mammography), there are no effective screening tests for OC—the few tests that can diagnose OC include transvaginal ultrasound, HE4 and CA125 determinations and ROMA algorithm determination, among others. However, these methods have limited sensitivity and specificity [9,11,12].

Today, modern oncology diagnostics focuses on early detection of cancer using minimally invasive methods such as determination of biomarkers from peripheral blood. Studies indicate the initial potential of biomarkers in the diagnosis of gynecologic cancers such as breast cancer [18,19], endometrial cancer [26,27] and cervical cancer [28]. Some reports also indicate the usefulness of various biomarkers in the diagnosis of OC [20,21,29]. Matrix metalloproteinases in particular have a high potential, and a large proportion of these enzymes have now been studied in patients with gynecologic cancers—among others in OC [20,21,24,25,29]. Therefore, the purpose of the present study was to further investigate the diagnostic utility of selected MMPs, MMP-2, MMP-3, MMP-11 and MMP-26, in patients with OC, benign ovarian lesions *(Serous cystadenomas*) and healthy women in comparison to routinely used markers, HE4 and CA125, and in comparison, to the ROMA algorithm. The MMPs were selected for the study after a literature review—they are currently molecules that have been poorly studied in OC and their blood concentrations have been determined in single studies or not previously performed.

MMP-2 expression is found in physiological ovarian tissue [25,30]. The presence of this enzyme is also found in OC samples (all histological subtypes)—both in the epithelial cells of the cancer and in the lining [30,31,32]. High expression of MMP-2 has been associated with certain clinical features of OC such as the propensity for recurrence and the presence of metastases [33]. Although most of the available studies have focused on MMP-2 expression, a single study has focused on blood MMP-2 concentrations in patients with OC. In our study, plasma MMP-2 concentrations were slightly higher in patients with benign lesions (BL) compared to OC patients, but this was not a statistically significant difference (BL—median: 211 ng/mL; OC—median: 203 ng/mL). At the same time, OC patients had higher levels of MMP-2 compared to healthy women (HL—median: 200.195 ng/mL). This disagrees with the work of Acar et al. [34] who showed higher lower MMP-2 concentrations in women with OC compared to healthy subjects (OC—median: 227.51 ng/mL; HL—median: 279.12 ng/mL). However, similar to our study, Acar et al. [34] found no significant differences between MMP-2 levels in OC and BL patients (BL—median: 247.01 ng/mL). The differences in the obtained results may be due to the size of the study group—in our study, we included 120 OC patients, while Acar et al. [34] included only 28. It should also be noted that our study involved plasma, while Acar et al. [34] performed the study on serum.

Similar to MMP-2, MMP-3 expression is found in physiological ovarian tissue [25,35]. mRNAs for MMP-3 have been found in OC samples, in all histological subtypes. Importantly, higher MMP-3 expression was observed with increasing tumor stage (according to FIGO) [36,37]. It is likely that high expression at higher FIGO stages may translate into high MMP-3 levels in the blood of OC patients as demonstrated by Cymbaluk-Płoska et al. [29]—patients with more advanced forms of the disease had higher serum MMP-3 levels. In addition, OC patients had higher MMP-3 concentrations than BL women (OC—median: 14.657 ng/mL; BL—median: 9.84 ng/mL). This does not agree with the results obtained by our team, as MMP-3 concentrations in OC patients and BL patients were similar to each other (OC—median: 9.32 ng/mL; BL—median: 9.84 ng/mL). These differences are likely due to differences between the groups selected for the study—the size of the groups and the greater variety of benign lesions. In addition, the study by Cymbaluk-Ploska et al. [29] was performed using multiplex fluorescent bead-based immunoassays. It is unfortunate that Cymbaluk-Ploska et al. [29] did not compare MMP-3 concentrations between OC and BL patients and healthy subjects; since this is the only such study available, we are unable to relate our results to other work—according to the results we obtained, healthy women had significantly lower MMP-3 concentrations (median: 7.97 ng/mL) than OC and BL patients. The collected data indicate the necessity of re-testing MMP-3 concentrations in OC patients in order to unequivocally confirm their diagnostic usefulness.

Unlike MMP-2 and MMP-3, MMP-11 is not expressed in normal ovarian tissue. The mRNA for MMP-11 is only found in OC samples (in all histologic subtypes) [25,37]. MMP-11 expression in OC patients was associated with higher FIGO stage [25,38]. To the best of our knowledge, we are the first team to determine MMP-11 levels in OC patients, so our results will be related to other cancer types. Our study shows that OC patients have higher MMP-11 concentrations (median: 1.50 ng/mL) compared to HL patients (0.24 ng/mL) and healthy subjects (0.51 ng/mL). In one of the few studies involved colorectal cancer patients, these patients also had higher levels of MMP-11 (median: 38.98 mg/mL) compared to healthy subjects (5.86 ng/mL). However, it should be noted that the study was conducted on serum [39].

Depending on the structure of the ovary, MMP-26 expression can be low or high [40]. As with other MMPs, this enzyme is expressed in OC tissue and mRNA levels increased with tumor progression (according to the FIGO classification) [40]. Again, our team was the first to determine MMP-26 in the plasma of OC patients—the highest MMP-26 levels were found in BL patients (median: 10.44 ng/mL) compared to OC (median: 9.33 ng/mL) and HL (median: 7.1625 ng/mL). Higher levels of MMP-26 in cancer patients compared to healthy individuals are also found in the course of other cancers such as breast cancer [41] and prostate cancer [42].

HE4 and CA125 are the most commonly used tumor markers in the diagnosis of OC [2,11,13]. In our study, we obtained higher concentrations of both markers in OC patients compared to BL patients and healthy women. This agrees with studies conducted by other research teams [21,43,44]. The results obtained in our study are similar to those of other research teams, which indicates that the methodology of our study was correct. We also used the ROMA algorithm in our study—according to our study, patients with OC had significantly higher ROMA algorithm values than patients with benign lesions. This also agrees with the results that have been obtained by other research teams [45,46,47,48].

As in other studies involving tumor markers, we also used diagnostic tools such as sensitivity, specificity, and negative and positive predictive value in our study. We obtained the highest values of the mentioned parameters for CA125, which is a routine marker; we also obtained very high values of diagnostic parameters for the ROMA algorithm. The diagnostic parameter values obtained in our study for CA125 (SE: 94.17%; SP: 98.00%; NPV: 87.50%; PPV: 99.12%) were higher than the results obtained by other research teams such as Będkowska et al. [20] (CA125—SE: 63%; SP: 91%; NPV: 82.1%; PPV: 79.7%) and Ławicki et al. [44] (CA125—SE: 67%; SP: 92%; NPV: 58%; PPV: 94%). In the case of the ROMA algorithm, the results we obtained (SE: 90.00% SP: 98.00% NPV: 99.08% PPV: 80.33%) are comparable to the work of other teams such as Cui et al. [49] (SE: 90.00% SP: 95.00% NPV: 95% PPV: 95%). Some research teams, in determining the usefulness of the ROMA algorithm, further divided patients by menopausal status; in our work, we did not make such a division due to the previously known diagnostic potential of this parameter.

In the case of MMPs, the best diagnostic parameter values were obtained for MMP-11 and MMP-26, but as mentioned they did not exceed those of routine markers and the ROMA algorithm. We are currently the first team to determine the diagnostic utility of MMP-11, so we are unable to relate our results to the work of other teams. In the case of MMP-26, our results relate in part to those obtained by Piskór et al. [41] who, however, found comparable values for the diagnostic parameters of MMP-26 compared to CA 15-3 (a routine marker in breast cancer) in breast cancer. In the case of MMP-2, we do not have studies to which we could relate our results, but in gastric cancer [50] and thyroid cancer [51], this marker acquired better diagnostic parameters than routine markers. Importantly, part of our study on MMP-3 overlaps with the results of Cymbaluk-Płoska et al. [29] where the SE (69%) and SP (67%) values acquired by this team were similar to our results (SE: 64.17%; SP:67%) and were lower than those of routine markers (HE4 and CA125).

The paper also determines the diagnostic power of the test—in single or combined analysis. The highest power of the test for single analysis is found for comparative markers and the ROMA algorithm. The AUC values obtained by our team for CA125 (AUC = 0.9918), HE4 (AUC = 0.9429) and the ROMA algorithm (AUC = 0.9336) were comparable to the results obtained for these parameters in a group of ovarian cancer patients. [20,21,29,45,48,49]. Depending on the new marker selected for the study, the AUC values for the comparative markers were lower or higher [20,21,45]. Among the MMPs tested, we found the highest AUC for MMP-26 (AUC = 0.7751); interestingly, Piskór et al. [41] in a group of breast cancer patients found an AUC for MMP-26 very similar to ours (0.7306). However, in these studies, a higher AUC is found for the routine marker. In the case of ovarian cancer patients, Cymbaluk-Płoska et al. [29] found an AUC value for MMP-3 of 0.76 which is also very similar to the AUC value obtained for MMP-3 in our study (AUC = 0.7453). Importantly, as in our study, the assay power for MMP-3 was lower than for the comparative markers.

Many studies determined the usefulness of modern markers and performed combined analyses with routine markers. However, in our work, we performed combined analyses of the MMPs studied with the ROMA algorithm. This is because the ROMA algorithm consists of more parameters (determination of two markers and determination of menopausal status) and, in addition, this algorithm provides a better predictive value than a single determination of HE4 and CA125 [12,45]. In addition, performing analyses of this type significantly reduced the number of reported results which, with the amount of data included in this work, was essential. Performing combined analyses of the ROMA algorithm with MMPs was associated with an increase in the power of the test. In contrast, we found the highest increase in diagnostic power of the assay after performing a combined analysis of MMP-2 + MMP-3 + MMP-11 + MMP-26 + ROMA (AUC = 0.9674). Importantly, increases in assay power after performing combined analyses were also obtained by other research teams working on new markers for ovarian cancer diagnosis [21,43].

Assessment of circulating tumor markers has high potential for early cancer diagnosis and can also be an adjunct test to existing testing methods. Although many currently studied markers from the metalloproteinase group, such as MMP-7 or MMP-9, have high potential in the diagnosis of ovarian cancer, the MMPs we studied did not show potential as independent markers. This is because, firstly, their concentrations were not always higher in the OC patient group, secondly, the diagnostic parameter values obtained for these markers were too low and did not exceed the diagnostic parameter values for routine markers, and thirdly, the power value of the assays for these markers was low compared to routine markers. Nevertheless, in our opinion, the MMPs tested show potential as an auxiliary test. According to our results, the combined analysis of MMPs with the ROMA algorithm increases the power of this assay. Although the greatest increase in the power of the assay was obtained for the combined analysis of all parameters, in our opinion, the greatest potential lies in the possibility of combining the ROMA algorithm with MMP-26. This is due to the fact that this will involve only one additional assay.

Unfortunate, our study has serious limitations. The first is the number of patients selected for the study. In the future, we would like to carry out similar determinations on a group of 500 patients with cancer, 250 patients with benign lesions and 250 healthy women; this will greatly increase the reliability of our results. An additional limitation of the work is the fact that we determined the MMPs under study by only one method (ELISA), so in a future study we plan to repeat our study with other methods as well.

In conclusion, we are the first team to study the significance of MMP-2, MMP-3, MMP-11 and MMP-26 as new markers of ovarian cancer in comparison not only to HE4 and CA125, but also to the ROMA algorithm. Through this, we hope that our study will contribute to the possibility of further research related to biochemical diagnosis of ovarian cancer and will be helpful to other researchers in this field.

## 4. Materials and Methods

Table 6 shows the characteristics of the groups participating in the study. A flowchart of the study is shown in Figure 9. The study included 120 patients with ovarian cancer. Based on World Health Organization guidelines and FIGO criteria, patients with ovarian cancer were assigned a clinical stage (I–IV) and histological classification (High-Grade Serous, Endometrioid, Mucinous and Clear–Cell Carcinoma). Histopathological evaluation was performed at the in-hospital diagnostic stage, the study material was an intraoperatively collected biopsy. Baseline examination, blood tests, ultrasound and X-rays were performed prior to the introduction of treatment. Patients did not receive chemotherapy or radiotherapy before the blood sample was taken. In addition, some patients underwent computed tomography (CT) or magnetic resonance imaging (MRI). Patients with renal failure were excluded from the study due to high HE4 levels. Written informed consent and medical histories were taken from all participants.

The control group consisted of 70 patients with benign ovarian tumor (Serous cystadenomas) and 50 healthy women. Patients with benign lesions were treated at the Department of Gynecology of the University Clinical Hospital in Bialystok from 2008 to 2012 and at the University Oncology Center of the University Clinical Hospital in Bialystok from 2019 to 2023. All patients underwent histopathological examination from biopsy material. Healthy women were volunteers included in the study after qualifying by a family physician and a gynecologist at the University Clinical Hospital in Bialystok. Volunteers selected for the study underwent annual preventive examinations (laboratory tests, cervical cytology, abdominal ultrasound), and patients with a clinical history of benign gynecological conditions were excluded. Before blood was drawn for testing, the volunteers were examined by a gynecologist, and an ultrasound was performed. Written consent to participate in the study and medical history were taken from all participants.

This study was performed in accordance with the Declaration of Helsinki. The protocol was authorized by the local Ethical Committee of the Medical University of Bialystok (committee approval numbers: APK.002.420.2021 (approval date 21 October 2021). All patients gave informed consent to participate in the study.

### 4.1. Biochemical Analyses

The study material was venous blood, collected for anticoagulant lithium heparin. Within an hour of collection, the blood was centrifuged at a speed of 1810× *g* for 10 min to obtain plasma. The plasma was then immediately separated and stored at −81 °C until assays were performed. Determination of selected MMPs was performed by enzyme-linked immunoassay (ELISA). The assay used kits from R&D Systems (MMP-2: Human MMP-2 DuoSet ELISA, Cat. No. DY902; MMP-3: Human Total MMP-3 DuoSet ELISA, Cat. No. DY513, Minneapolis, MN, USA), ElabScience (MMP-11: Human MMP-11(Matrix Metalloproteinase 11) ELISA Kit, Cat. No. E-EL-H1443, Wuhan, China) and Abbkine (MMP-26: Human Matrix metalloproteinase-26 (MMP26) ELISA Kit, Cat. No. KTE61590, Georgia, GA, USA). The assay procedure was performed according to the manufacturer’s recommendations included with each kit.

Standards and samples were made in duplicate. The precision of the kits was set by the manufacturer—MMP-2: 3.8% (Intra-assay), 6.6% (Inter-assay); MMP-3: 6.1% (Intra-assay), 7% (Inter-assay); MMP-11: 5.6% (Intra-assay), 5% (Inter-assay); and MMP-26: <9% (Intra-assay), <11% (Inter-assay). Plates were read by using a wavelength of 450 nm and a correction set at 540 nm.

Routine markers HE4 and CA125 were measured by chemiluminescent microparticle immunoassay (CMIA) (Abbott, Chicago, IL, USA) according to the manufacturer’s protocols on a Cobas e411 instrument using reagents from ROCHE Diagnostics (Roche Elecsys HE4 and Roche Elecsys CA125 II).

### 4.2. ROMA Calculations

The ROMA algorithm was calculated based on the work of Kumar et al. [24] and Anton et al. [25]. The predictive index (PI) values were calculated as follows, depending on the menopausal status of the patients:Premenopausal woman: (PI) = −12.0 + 2.38 × LN (HE4) + 0.0626 × LN (CA125).Postmenopausal woman: (PI) = −8.09 + 1.04 × LN (HE4) + 0.732 × LN (CA125).

The natural logarithm (LN) formula was used in the calculations.

For premenopausal woman, a value of ≥11.4% was assumed for high risk of ovarian cancer, while a low risk of <11.4% was assumed.

For a postmenopausal woman, a value of ≥29.9% was assumed for a high risk of ovarian cancer, while a low risk of <29.9% was assumed.

The ROMA algorithm was not determined in patients previously treated for cancer and undergoing chemotherapy.

### 4.3. Statistical Analysis

Statistical analysis was performed using the statistical program PQStat Software (v.1.8.4.162, Poznań, Poland); graphical processing was performed using GraphPad Prism Software (v. 9.1.1 (225), San Diego, CA, USA). Normality of distribution was evaluated using the Shapiro–Wilk test. Further analyses were carried out by using non-parametric tests—Kruskal–Wallis’s test with the Conover–Iman post hoc test for performing group comparisons, and the Spearman rank correlation test for determining correlations between parameters. Using the ROC curve, an analysis of the diagnostic reliability and diagnostic power of the tests was performed with the optimal cut-off point determined for MMP-2 (219 ng/mL), MMP-3 (8.81 ng/mL), MMP-11 (0.92 ng/mL), MMP-26 (8.23 ng/mL), HE4 (66.3 U/mL) and CA125 (30.5 U/mL). Comparisons of ROC curves were made using AUCs determined by the nonparametric De Long method. The figures were created using Graph Pad Prism 5 software (GraphPad Software, La Jolla, CA, USA).

## 5. Conclusions

MMP-2, MMP-3, MMP-11 and MMP-26 have not shown potential as stand-alone biochemical markers in ovarian cancer diagnosis. However, they may be used as auxiliary markers for the ROMA algorithm in the future.

## Figures and Tables

**Figure 1 ijms-25-06265-f001:**
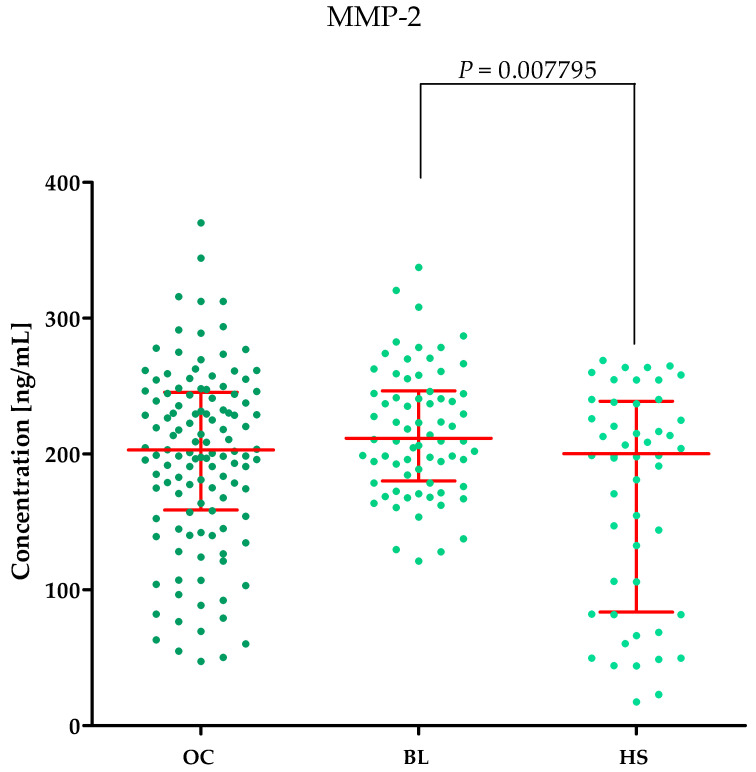
MMP-2 plasma concentrations (with marked median and interquartile range) in all tested groups: patients with ovarian cancer (OC), begin lesion (BL) and healthy subject (HS). MMP-2 concentrations were obtained by ELISA method.

**Figure 2 ijms-25-06265-f002:**
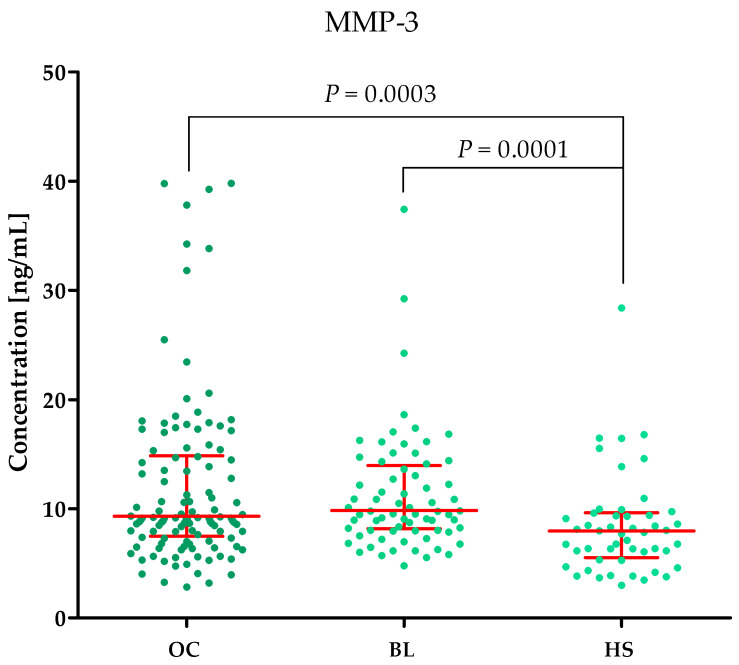
MMP-3 plasma concentrations (with marked median and interquartile range) in all tested groups: patients with ovarian cancer (OC), begin lesion (BL) and healthy subject (HS). MMP-3 concentrations were obtained by ELISA method.

**Figure 3 ijms-25-06265-f003:**
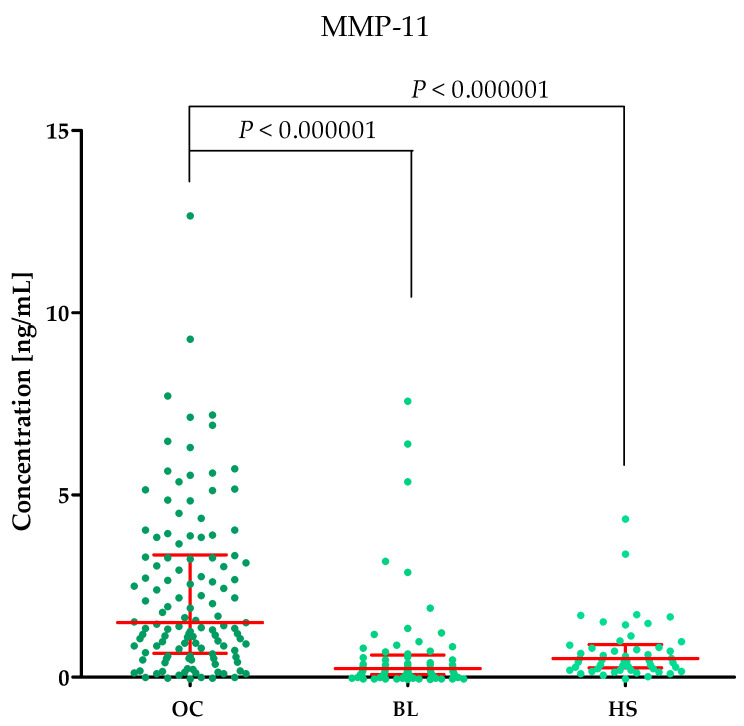
MMP-11 plasma concentrations (with marked median and interquartile range) in all tested groups: patients with ovarian cancer (OC), begin lesion (BL) and healthy subject (HS). MMP-11 concentrations were obtained by ELISA method.

**Figure 4 ijms-25-06265-f004:**
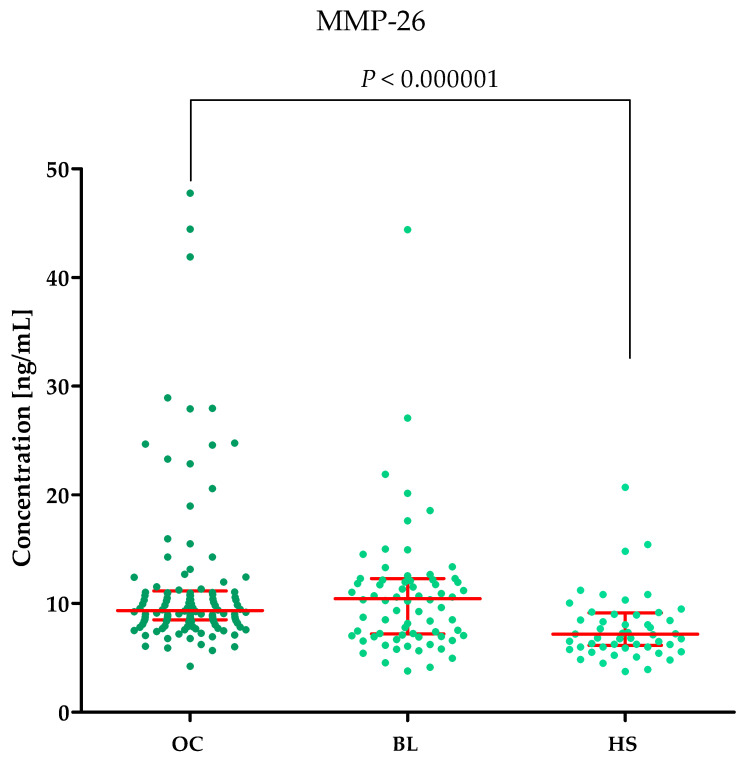
MMP-26 plasma concentrations (with marked median and interquartile range) in all tested groups: patients with ovarian cancer (OC), begin lesion (BL) and healthy subject (HS). MMP-26 concentrations were obtained by ELISA method.

**Figure 5 ijms-25-06265-f005:**
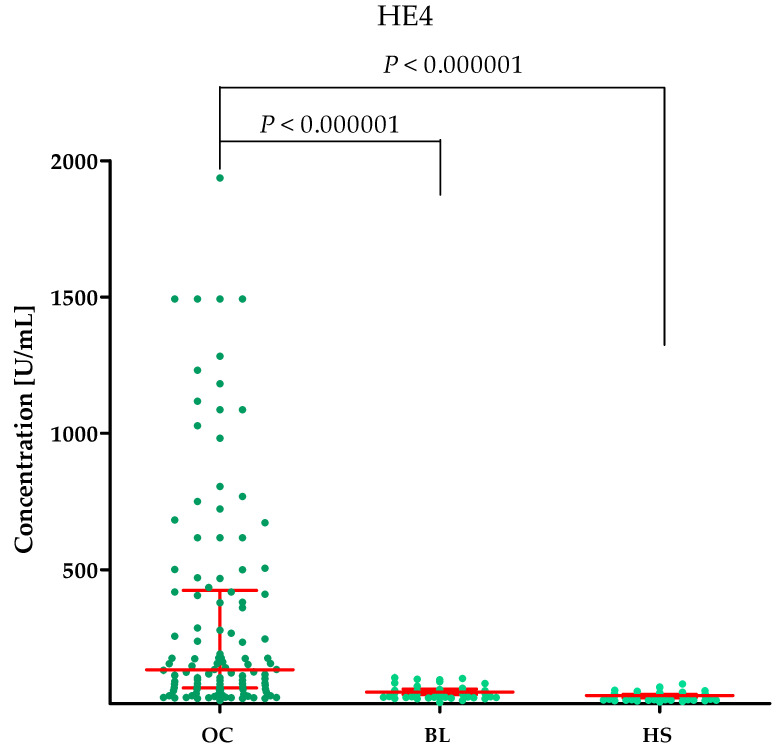
HE4 plasma concentrations (with marked median and interquartile range) in all tested groups: patients with ovarian cancer (OC), begin lesion (BL) and healthy subject (HS). HE4 concentrations were obtained by the CMIA method.

**Figure 6 ijms-25-06265-f006:**
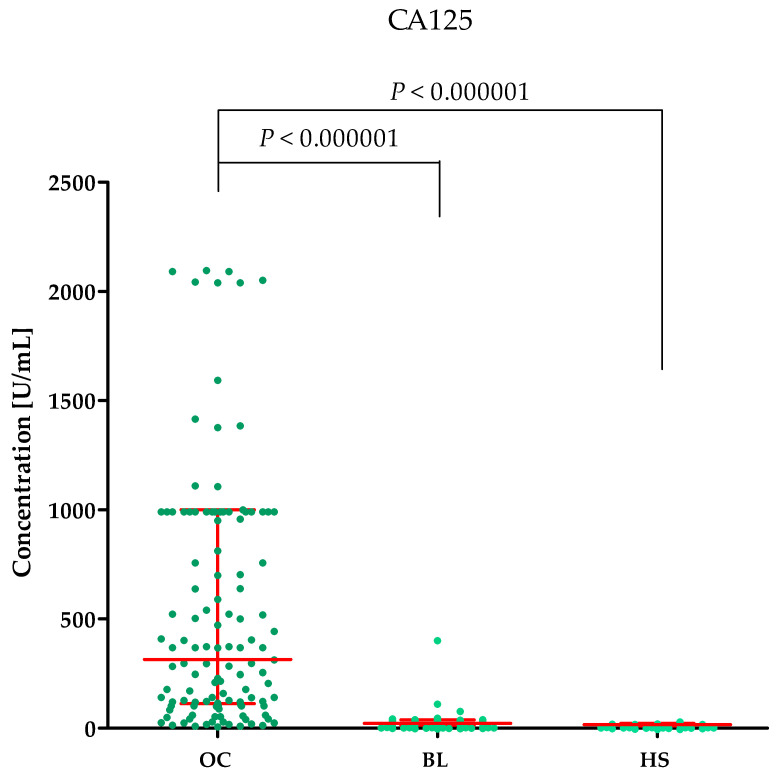
CA125 plasma concentrations (with marked median and interquartile range) in all tested groups: patients with ovarian cancer (OC), begin lesion (BL) and healthy subject (HS). CA125 concentrations were obtained by the CMIA method.

**Figure 7 ijms-25-06265-f007:**
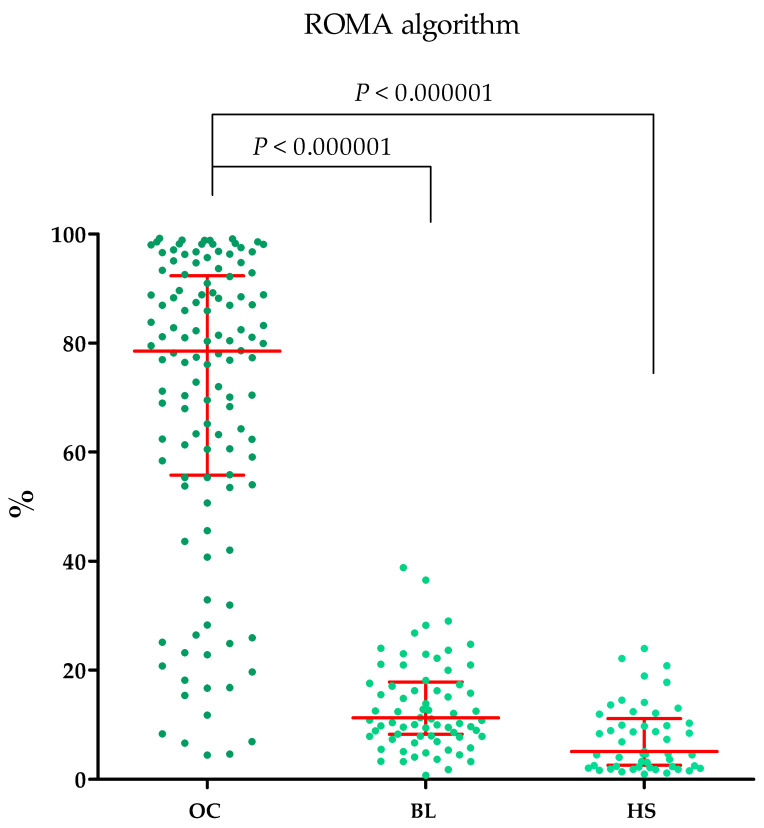
ROMA algorithm values expressed in percentages (with marked median and interquartile range) in patients with ovarian cancer (OC) and begin lesion (BL).

**Figure 8 ijms-25-06265-f008:**
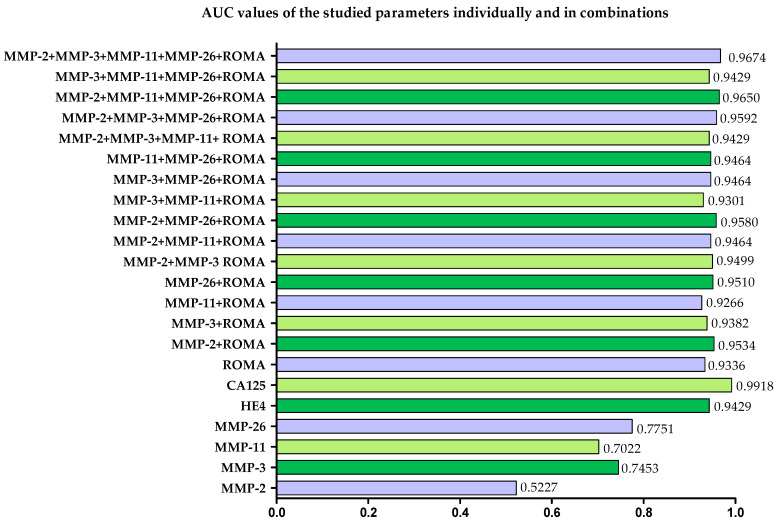
AUC values obtained for all parameters and combinations between parameters.

**Figure 9 ijms-25-06265-f009:**
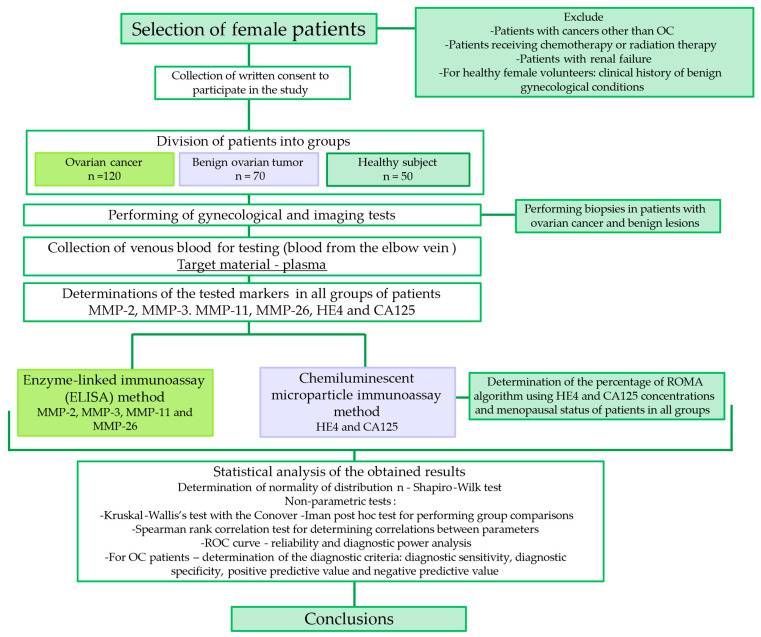
Flowchart of the study course.

**Table 1 ijms-25-06265-t001:** Plasma concentrations of studied parameters MMP-2, MMP-3, MMP-11 and MMP-26, comparative markers CA125 and HE4 and value of the ROMA algorithm expressed in percentages in ovarian cancer patients (OC), begin lesion patients (BL) and healthy subject (HS). MMP concentrations were obtained by enzyme-linked immunoassay (ELISA), while HE4 and CA125 concentrations were obtained by chemiluminescent microparticle immunoassay method (CMIA).

Ovarian Cancer							
	MMP-2 (ng/mL)	MMP-3 (ng/mL)	MMP-11 (ng/mL)	MMP-26 (ng/mL)	HE4 (U/mL)	CA125 (U/mL)	ROMA (%)
**Median**	203.00	9.32	1.50	9.33	132.95	313.55	78.515
**Min–max range**	48.60–371.60	3.03–40.00	0.02–17.26	4.39–47.95	28.90–1944.20	17.10–4113.10	4.86–99.59
**IQR**	159.225–245.125	7.5025–14.7425	0.685–3.345	8.515–11.125	67.70–420.325	112.85–1000.00	55.75–91.70
**Benign lesions**							
**Median**	211.50	9.84	0.24	10.44	52.50	11.253	11.25
**Min–max range**	122.50–338.88	4.97–37.65	0.00–7.64	3.96–44.6	21.40–112.3	5.80–410.3	1.13–39.21
**IQR**	181.50–245.97	8.1825–13.69	0.085–0.585	7.225–12.24	14.75–37.40	43.75–62.675	8.29–17.69
**Healthy subjects**							
**Median**	200.195	7.97	0.51	7.1625	39.505	16.9	5.09
**Min–max range**	19.00–270.40	3.18–28.59	0.02–4.40	3.92–20.89	23.52–89.37	4.24–39.94	1.36–24.36
**IQR**	89.425–235.75	5.7125–9.5775	0.27–0.875	6.1563–8.9988	11.78–21.745	31.172–43.05	2.65–10.63

IQR—interquartile range.

**Table 2 ijms-25-06265-t002:** The Spearman’s rank correlation test for MMP-2, MMP-3, MMP-11, MMP-26, HE4 and CA125 in tested groups.

Tested Correlations														
	MMP-2	MMP-2	MMP-2	MMP-3	MMP-3	MMP-11	MMP-2	MMP-2	MMP-3	MMP-3	MMP-11	MMP-11	MMP-26	MMP-26
	vs.													
	MMP-3	MMP-11	MMP-26	MMP-11	MMP-26	MMP-26	CA125	HE4	CA125	HE4	CA125	HE4	CA125	HE4
**Ovarian cancer**														
r	0.0793	−0.0186	−0.1139	0.0992	−0.0400	−0.0496	0.0263	−0.0696	0.0790	0.0765	0.0730	0.2220	−0.0538	0.0176
*p*	0.3890	0.8404	0.2155	0.2812	0.6643	0.5906	0.7752	0.4497	0.3911	0.4060	0.4282	0.0148	0.5598	0.8490
**Benign lesions**														
r	0.1092	0.0321	−0.0201	−0.1332	−0.0240	−0.2241	0.1113	0.0241	0.1211	−0.0929	−0.0862	0.0129	0.0185	−0.0739
*p*	0.3683	0.7920	0.8690	0.2717	0.8435	0.0622	0.3590	0.8433	0.3179	0.4444	0.4782	0.9157	0.8790	0.5430
**Healthy subjects**														
r	0.0643	−0.3445	−0.0038	0.1806	−0.0728	0.0120	−0.1601	−0.0250	0.0971	0.1038	0.2702	0.0058	−0.0079	0.0735
*p*	0.6571	0.0143	0.9791	0.2096	0.6156	0.9343	0.2666	0.8629	0.5022	0.4731	0.0577	0.9683	0.9564	0.6121

Red color indicates statistically significant correlations of tested parameters.

**Table 3 ijms-25-06265-t003:** The Spearman’s rank correlation test for MMP-2, MMP-3, MMP-11, MMP-26 and ROMA algorithm in tested groups.

Tested Correlations				
	MMP (ng/mL)	MMP (ng/mL)	MMP1 (ng/mL)	MMP6 (ng/mL)
	vs.			
	ROMA %	ROMA %	ROMA %	ROMA %
**Ovarian cancer**				
r	0.0365	0.1057	0.2026	0.0510
*p*	0.6923	0.2508	0.0265	0.5802
**Benign lesions**				
r	0.0938	0.0906	0.0153	0.0589
*p*	0.4397	0.4555	0.8997	0.6279
**Healthy subjects**				
r	0.1329	0.1525	0.0916	0.0656
*p*	0.3574	0.2903	0.5270	0.6506

Red color indicates statistically significant correlations of tested parameters.

**Table 4 ijms-25-06265-t004:** Diagnostic utility of tested parameters individually and in combination.

Parameter	SE	SP	PPV	NPV
**MMP-2**	42.50	68.00	76.12	33.01
**MMP-3**	64.17	68.00	82.80	44.16
**MMP-11**	70.83	76.00	87.63	52.05
**MMP-26**	78.33	68.00	85.45	56.67
**HE4**	76.67	92.00	95.83	62.16
**CA125**	94.17	98.00	99.12	87.50
**ROMA**	90.00	98.00	99.08	80.33
**MMP-2 + ROMA**	91.67	98.00	99.10	83.05
**MMP-3 + ROMA**	90.83	100.00	100.00	81.97
**MMP-11 + ROMA**	92.50	94.00	97.37	83.93
**MMP-26 + ROMA**	94.17	96.00	98.26	87.27
**MMP-2 + MMP-3 + ROMA**	92.50	96.00	98.23	84.21
**MMP-2 + MMP-11 + ROMA**	92.50	94.00	97.37	83.93
**MMP-2 + MMP-26 + ROMA**	93.33	98.00	99.12	85.96
**MMP-3 + MMP-11 + ROMA**	92.50	94.00	97.37	83.93
**MMP-3 + MMP-26 + ROMA**	95.83	96.00	98.29	90.57
**MMP-11 + MMP-26 + ROMA**	93.33	98.00	99.12	85.96
**MMP-2 + MMP-3 + MMP-11 + ROMA**	90.83	98.00	99.09	81.67
**MMP-2 + MMP-3 + MMP-26 + ROMA**	94.17	100.00	100.00	87.72
**MMP-2 + MMP-11 + MMP-26 + ROMA**	95.83	96.00	98.29	90.57
**MMP-3 + MMP-11 + MMP-26 + ROMA**	93.33	98.00	99.12	85.96
**MMP-2 + MMP-3 + MMP-11 + MMP-26 + ROMA**	95.83	96.00	98.29	90.57

**Table 5 ijms-25-06265-t005:** Characteristics of ROC curve for tested parameters individually and in combination in OC patients.

Parameter	AUC	SE_AUC_	95% CI	*p* (AUC = 0.5)
**MMP-2**	0.5227	0.0768	0.3722–0.6733	0.7659
**MMP-3**	0.7453	0.0647	0.6186–0.8721	0.0013
**MMP-11**	0.7022	0.0747	0.5558–0.8487	0.0081
**MMP-26**	0.7751	0.0629	0.6518–0.8983	0.0003
**HE4**	0.9429	0.0268	0.8904–0.9954	<0.000001
**CA125**	0.9918	0.0078	0.9766–1.0000	<0.000001
**ROM** **A**	0.9336	0.0299	0.8749–0.9923	<0.000001
**MMP-2 + ROMA**	0.9534	0.0242	0.9060–1.0000	<0.000001
**MMP-3 + ROMA**	0.9382	0.0298	0.8799–0.9966	<0.000001
**MMP-11 + ROMA**	0.9266	0.0366	0.8548–0.9984	<0.000001
**MMP-26 + ROMA**	0.9510	0.0248	0.9024–0.9997	<0.000001
**MMP-2 + MMP-3 + ROMA**	0.9499	0.0249	0.9012–0.9986	<0.000001
**MMP-2 + MMP-11 + ROMA**	0.9464	0.0280	0.8915–1.0000	<0.000001
**MMP-2 + MMP-26 + ROMA**	0.9580	0.0215	0.9159–1.0000	<0.000001
**MMP-3 + MMP-11 + ROMA**	0.9301	0.0365	0.8586–1.0000	<0.000001
**MMP-3 + MMP-26 + ROMA**	0.9464	0.0271	0.8933–0.9995	<0.000001
**MMP-11 + MMP-26 + ROMA**	0.9464	0.0336	0.8805–1.0000	<0.000001
**MMP-2 + MMP-3 + MMP-11 + ROMA**	0.9429	0.0301	0.8839–1.0000	<0.000001
**MMP-2 + MMP-3 + MMP-26 + ROMA**	0.9592	0.0214	0.9172–1.0000	<0.000001
**MMP-2 + MMP-11 + MMP-26 + ROMA**	0.9650	0.0219	0.9221–1.0000	<0.000001
**MMP-3 + MMP-11 + MMP-26 + ROMA**	0.9429	0.0360	0.8723–1.0000	<0.000001
**MMP-2 + MMP-3 + MMP-11 + MMP-26 + ROMA**	0.9674	0.0214	0.9255–1.0000	<0.000001

Red color indicates statistically significant correlations of tested parameters.

**Table 6 ijms-25-06265-t006:** Characteristic of examined groups: ovarian cancer (histological diagnosis and stage, menopausal status and age), benign lesions (histological diagnosis, menopausal status and age) and healthy subjects (menopausal status and age).

Ovarian Cancer	
**Number of patients**	120 (100%)
**Histopathological diagnosis**	(1) High-Grade Serous Carcinoma—87 (72.5%) (2) Endometrioid Carcinoma—30 (25%) (3) Mucinous Carcinoma—2 (1.6%) (4) Clear-Cell Carcinoma—1 (0.9%)
**Tumor stage**	(1) Stage I—17 (14.2%) IA—4 IB—5 IC—8 (2) Stage II—29 (24.2%) IIA—10 IIB—11 IIC—8 (3) Stage III—38 (31.6%) IIIA—16 IIIB—14 IIIC—8 (4) Stage IV—36 (30%)
**Menopausal status**	(1) Pre-menopause—26 (21.7%) (2) Post-menopause—94 (78.3%)
**Median age (range)**	59.5 (22–80)
**Control group**	
**Benign lesions**	
**Number of patients**	70 (100%)
**Histopathological diagnosis**	(1) *Serous cystadenomas*—70 (100%)
**Menopausal status**	(1) Pre-menopause—31 (44.3%) (2) Post-menopause—39 (55.7%)
**Median age (range)**	51 (16–80)
**Healthy subjects**	
**Number of patients**	50 (100%)
**Menopausal status**	(1) Post-menopause—50 (100%)
**Median age (range)**	50 (45–65)

## Data Availability

Data is contained within the article.

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
