# Peer review of "Diagnostic Utility of Selected Matrix Metalloproteinases (MMP-2, MMP-3, MMP-11, MMP-26), HE4, CA125 and ROMA Algorithm in Diagnosis of Ovarian Cancer"

_ijms, 2024, doi:10.3390/ijms25116265_

Round 1
Reviewer 1 Report
Comments and Suggestions for Authors
In this study, the authors evaluated the diagnostic utility of MMPs and the combination of MMPs with the POMA algorithm. Although it’s an interesting study, I have the following concerns about this study.
1. It seems that CA125 alone could achieve a 94.17% sensitivity, a 98.00% specificity, and an AUC of 0.9918 to diagnose OC. Although the combination with MMPs improved the sensitivity to 95.83%, the specificity decreased to 96.00%, and the AUC decreased to 0.9674. This needs to be explained. Because testing of the MMPs combination may cost more than testing of CA125. Also, it’s unclear which control (HS, BL, or both) you used to evaluate the diagnostic utility of parameters in Table 4 and Table 5.
2. SE values in Table 4 are dramatically inconsistent with those in Table 5. How do you explain the big difference?
3. MMP-2 showed no difference between OC and HS (Figure 1) with an AUC of 0.5227 (95% CI: 0.3722-0.6733). Why should it still be included in the combination? Based on your reference 18, you have already evaluated the diagnostic utility of MMP-2 in OC and it showed no significant difference. Why the MMP-2 was included in this study again?
4. Figure 4 OC vs. BL showed a significant difference. However, lines 140-142 described no statistical difference.
5. Result 2.1.2 is unclear. If the PI for ROMA was calculated as mentioned in materials and methods, why the values of 90.32% and 100% are considered as low risk of OC?
6. Result 2.2 should claim the purpose of doing the correlations. Have you checked the expression of MMPs in different menopausal statuses?
7. It’s unclear why the authors selected the four MMPs. And how did the authors define the cut-off values for those MMPs?
8. The abstract can be improved. For example, in lines 29-32, “performing …increased ….., performing …increased” and typos (algoritm).
9. The way to report P values should be the same in this manuscript. P should be italicized and capitalized.
10. Lines 79 and 89 mentioned matrix metalloproteinases (MMPs) twice. Line 90 lacks references. Also, you have two tables named Table 2. The first Table 2 shows the Median, Range, and IQR (may need a footnote to give the IQR full name). However, the decimal for each value is not at the same place.
Author Response
In this study, the authors evaluated the diagnostic utility of MMPs and the combination of MMPs with the POMA algorithm. Although it’s an interesting study, I have the following concerns about this study.
Dear Reviewer.
We would like to thank You very much, for performing a thorough review of our manuscript entitled "Diagnostic utility of selected matrix metalloproteinases (MMP-2, MMP-3, MMP-11, MMP-26), HE4, CA125 and ROMA algorithm in diagnosis of ovarian cancer" and for all Your useful suggestions and spotting errors and typos. Any changes to the manuscript have been highlighted in green and responses to the reviewer have been highlighted in green and cursive.
- It seems that CA125 alone could achieve a 94.17% sensitivity, a 98.00% specificity, and an AUC of 0.9918 to diagnose OC. Although the combination with MMPs improved the sensitivity to 95.83%, the specificity decreased to 96.00%, and the AUC decreased to 0.9674. This needs to be explained. Because testing of the MMPs combination may cost more than testing of CA125.
Thank You for Your valid comment.
We agree that performing a combination of MMPs+ROMA was associated with an increase in sensitivity and a decrease in specificity and AUC. Ideally, after performing such an analysis, we would observe an increase in all parameters, however, it is unfortunate that in our results we did not get such a relationship. This is most likely due to the fact that, the ROMA algorithm consists of two markers simultaneously. Both CA125 and HE4 have a number of important limitations and their values can be affected by both physiological and pathological conditions in the female body. In addition, this effect may also be related to the size of the study groups. In the future, we plan to repeat this study on a larger group of patients and compare the obtained results with the present work.
Currently, MMPs are only determined in an experimental framework and cannot be determined by routine methods in the laboratory, so it is difficult for us to assess at this point how much such tests could cost. In addition, we have not performed analysis of individual MMPs with CA125 or MMPs with HE4 by which we are unable to say whether such analysis increases the values of diagnostic parameters, however, such tests may have high potential due to the fact mentioned earlier that CA125 and HE4 titers are influenced by many factors and that CA125 is not organ-specific (for example, it is also used in the diagnosis of emdometrial cancer). HE4 is also not exclusively limited to ovarian cancer. However, we believe that this does not detract from the potential of our study and it can still be useful in the field of biochemical diagnosis of ovarian cancer.
Also, it’s unclear which control (HS, BL, or both) You used to evaluate the diagnostic utility of parameters in Table 4 and Table 5.
Thank You for Your comment.
manuscript is another in a series of papers on modern tumor markers in the diagnosis of cancer of the reproductive organs. The evaluation of diagnostic parameters in this manuscript and in each previous paper was performed after consultation with a biostatistician and related to a group of healthy patients.
- SE values in Table 4 are dramatically inconsistent with those in Table 5. How do You explain the big difference?
Thank You for Your comment.
This is a big shortcoming on our part, because in Table 5, the abbreviation SE does not mean sensitivity only AUC standard error. We have modified this abbreviation accordingly. After the correction, the abbreviation SEAUC stands for standard error AUC. All the changes have been marked in green in the manuscript.
- MMP-2 showed no difference between OC and HS (Figure 1) with an AUC of 0.5227 (95% CI: 0.3722-0.6733). Why should it still be included in the combination? Based on You r reference 18, You have already evaluated the diagnostic utility of MMP-2 in OC and it showed no significant difference. Why the MMP-2 was included in this study again?
Thank You for Your comment.
First, we are aware that MMP-2 as a stand-alone marker has low values of diagnostic parameters that do not exceed the values of routine markers and the ROMA algorithm, however, in our opinion, this does not preclude its use in combined analyses. Performing combined analyses with MMP-2 was associated with an increase in the values of some diagnostic parameters, which indicates MMP-2's potential as an auxiliary test. Again, we intend to perform similar analyses with data from a larger number of patients to make the potential of MMP-2 unequivocal.
Secondly, when deciding on the selection of MMPs for the present study, we had in mind the work of Będkowska et al. [18] - it should be noted, the cancer patients selected by this team had different histo-pathological diagnoses than the patients in our study and, in addition, our control group of patients with benign lesions consisted exclusively of patients with serous cysts. Therefore, we wanted to confirm, the data obtained by Bêdkowska et al. [18] on our group of patients and also perform combined analyses of MMP-2 with the ROMA algorithm, which the aforementioned team did not do.
- Figure 4 OC vs. BL showed a significant difference. However, lines 140-142 described no statistical difference.
Thank You for Your comment.
This is an error in the figure. In our study we did not show, statistically significant differences between OC and BL for MMP-26. We are very sorry for the mistake and thank You for noticing the error. The figure has been corrected accordingly
- Result 2.1.2 is unclear. If the PI for ROMA was calculated as mentioned in materials and methods, why the values of 90.32% and 100% are considered as low risk of OC?
Thank You for Your comment.
The values given in parentheses are not the values of the ROMA algorithm extracted for patients and the number of women with high or low risk. We agree that such a description of the data can be misleading, so we have modified the passage accordingly. The changes have been highlighted in green.
- Result 2.2 should claim the purpose of doing the correlations. Have You checked the expression of MMPs in different menopausal statuses?
Thank You for Your comment.
Firstly, correlations by Spearman's method were calculated to assess the presence of potential relationships between the parameters studied - this was mentioned in the description of the methodology for performing statistical analyses.
Second, due to the large amount of data processed , we did not perform an analysis of the expression and concentrations of MMPs according to menopausal status, however, we plan to do so in future studies. We did, however, analyze the concentrations of MMPs depending on the stage of ovarian cancer according to FIGO, but we did not show any statistical significance for all the MMPs studied, so we did not show these results in our paper.
- It’s unclear why the authors selected the four MMPs. And how did the authors define the cut-off values for those MMPs?
Thank You for Your comment.
First of all, the selection of specific MMPs for the study was based on a thorough analysis of the literature, and we also relied on the thorough review work done by the first author of this manuscript [doi:10.2147/CMAR.S385658]. The selection of specific MMPs for the paper is not coincidental and is due to the lack of studies of these compounds in serum or the limited number of them. We agree that studies on the potential of MMP-2 have already been done by the team of Będkowska et al [18] however, we wanted to replicate this study using our group of patients and to unambiguously determine the potential of MMP-2 in the diagnosis of ovarian cancer.
Secondly, the cutoff points were determined using the " closest distance to corner method", this analysis was performed by a professional biostatistician and was selected for the study based on our previous experience.
- The abstract can be improved. For example, in lines 29-32, “performing …increased ….., performing …increased” and typos (algoritm).
Thank You for Your comment.
The abstract has been revised accordingly to remove repeated words and typos. Any changes to the manuscript have been highlighted in green.
- The way to report P values should be the same in this manuscript. P should be italicized and capitalized.
Thank You for Your comment, all P designations in the manuscript will be in capital letters and italics. Any changes have been highlighted in green.
- Lines 79 and 89 mentioned matrix metalloproteinases (MMPs) twice. Line 90 lacks references. Also, You have two tables named Table 2. The first Table 2 shows the Median, Range, and IQR (may need a footnote to give the IQR full name). However, the decimal for each value is not at the same place.
First, thank You for all Your comments. We have made corrections in lines 79 and 89 - we have removed the expanded name of MMPs so that the abbreviation is expanded at this point only in line 79.
Secondly, we have corrected the numbering of the tables. Thank You for noticing the mistake in the numbering of the tables.
Third, we have expanded the abbreviation - IQR (interquartile range), above the table.
Fourth, we set the decimal results in the tables in the right way.
Again, we thank the reviewer for all the guidance and corrections. It is our hope that the manuscript, after revision, will meet the reviewer's expectations and be published in the International Journal of Molecular Sciences.
Best regards,
prof. dr hab. Sławomir Ławicki
also, on behalf of all authors

Reviewer 2 Report
Comments and Suggestions for Authors
The authors evaluated the usability of matrix metalloproteinases (MMPs) MMP-2, MMP-3, MMP-11 and MMP-26 as new markers of ovarian cancer. The study involved 120 patients with ovarian cancer, 70 patients with benign lesions and 50 healthy women. Alongside testing the sensitivity of MMP in ovarian cancer patients, both CA125 and H24 levels were also determined. Based on the results the authors conclude that MMPs as a standalone biomarker for ovarian cancer do not show potential.
Minor Revision:
1. Line 322: The line reads with both higher & lower, please make the necessary change.
2. Table 5: Can be improved for better readability?
3. It’s not clear on how the samples were collected for healthy volunteers. Please clarify.
4. The is a lot of variability in the MMP-2, MMP-3, MMP-11 and MMP-26 data among the three groups. Is there any reasoning for such high variability in the dataset? Would the variability reduce if a specific age group was chosen for the study?
5. Addition of bioanalysis details of the MMP quantitation would help the field to conduct such studies.
Author Response
The authors evaluated the usability of matrix metalloproteinases (MMPs) MMP-2, MMP-3, MMP-11 and MMP-26 as new markers of ovarian cancer. The study involved 120 patients with ovarian cancer, 70 patients with benign lesions and 50 healthy women. Alongside testing the sensitivity of MMP in ovarian cancer patients, both CA125 and H24 levels were also determined. Based on the results the authors conclude that MMPs as a standalone biomarker for ovarian cancer do not show potential.
Dear Reviewer.
We would like to sincerely thank You for your input and effort in reviewing our work – “Diagnostic utility of selected matrix metalloproteinases (MMP-2, MMP-3, MMP-11, MMP-26), HE4, CA125 and ROMA algorithm in diagnosis of ovarian cancer” and for Your valuable suggestions. We will try to follow all suggestions received in this review. Responses will be indicated in green italics and any changes to the manuscript in light green.
Minor Revision:
- Line 322: The line reads with both higher & lower, please make the necessary change.
Thank You for Your comment.
We have made an improvement in the manuscript. The change has been highlighted in green.
- Table 5: Can be improved for better readability?
Thank you for Your comment.
Table 5 has been redesigned accordingly. Since Table 4 has a similar layout, we also decided to redo this table in the same way as Table 5.
- It’s not clear on how the samples were collected for healthy volunteers. Please clarify.
Thank You for Your comment.
The blood sample from healthy volunteers was collected in the same way as from patients with OC or BL. The blood was collected using the vacuum method from the elbow vein. This is described in more detail in the study methodology. Healthy volunteers were examined by a gynecologist to rule out reproductive tract disorders prior to collection.
- The is a lot of variability in the MMP-2, MMP-3, MMP-11 and MMP-26 data among the three groups. Is there any reasoning for such high variability in the dataset? Would the variability reduce if a specific age group was chosen for the study?
Again, thank You for Your comment.
We realize that there is a fair amount of variability in the data between the three groups, however, at this point we do not have a clear rationale for the existence of such variability. Age, indeed, may be one of the many reasons for this distribution of data. In our study, we selected patients within a certain age range (a higher percentage of postmenopausal patients), which was mainly due to the trend of OC in older women. In our next studies, once we have collected enough material, we plan to conduct experiments by age group - pre-menopausal and post-menopausal.
- Addition of bioanalysis details of the MMP quantitation would help the field to conduct such studies.
Thank you for your comment.
The statistics for our study, was performed by a professional biostatistician. The tests he selected are adequately described in the research methodology. In our opinion, the description of the statistical methodology is adequate and clarifies any issues related to the topic.
Again, we thank the reviewer for all the guidance and corrections. It is our hope that the manuscript, after revision, will meet the reviewer's expectations and be published in the International Journal of Molecular Sciences.
Best regards,
prof. dr hab. Sławomir Ławicki
also, on behalf of all authors

Reviewer 3 Report
Comments and Suggestions for Authors
This study evaluated the diagnostic utility of various MMPs in diagnosis of ovarian cancer. In the introduction section (which is too long) the authors state that ‘the outstanding potential is characterized by extracellular matrix metalloproteinases (MMPs)’ but do not provide appropriate reference. It is not mentioned in the introduction section that no effective screening strategies exist for the prevention of OC. It is not clear whether this is a prospective or retrospective study. When were the patients included? What were the exclusion criteria for the study group? It is not clear when the blood for biochemical analysis was drawn? Was it before or after diagnosis?
Author Response
Reviewer 3.
This study evaluated the diagnostic utility of various MMPs in diagnosis of ovarian cancer. In the introduction section (which is too long) the authors state that ‘the outstanding potential is characterized by extracellular matrix metalloproteinases (MMPs)’ but do not provide appropriate reference.
Dear reviewer.
We sincerely thank you for such a positive review of our paper - „Diagnostic utility of selected matrix metalloproteinases (MMP-2, MMP-3, MMP-11, MMP-26), HE4, CA125 and ROMA algorithm in diagnosis of ovarian cancer”. We will try our best to answer each reviewer's question accurately and honestly. Answers will be written in green italics, while we will mark changes in the manuscript in light green.
First, Thank You for your comment.
We apologize for the missing reference, the error has been corrected and the corresponding reference added to the manuscript (Kicman, A.; Niczyporuk, M.; Kulesza, M.; Motyka, J.; Ławicki, S. Utility of Matrix Metalloproteinases in the Diagnosis, Monitoring and Prognosis of Ovarian Cancer Patients. Cancer Manag Res 2022, 14, 3359-3382, doi: 10.2147/CMAR.S385658.). At the same time, with all due respect, we do not consider the introduction to be too long; in our opinion, the introduction in this form best introduces the reader to the issues presented in our manuscript.
-It is not mentioned in the introduction section that no effective screening strategies exist for the prevention of OC
Thank You for Your comment.
We would like to point out that in the introduction we took into account the fact that in the course of ovarian cancer medicine does not dissonate screening tests. We refer You to lines 61-70 where we described the above issue.
-It is not clear whether this is a prospective or retrospective study.
Thank You for Your comment.
Our study is a retrospective study because it refers to existing medical records.
-When were the patients included?
Thank You for Your comment.
Patients from all groups were included in the study in two-time rounds. The first round of patients was from 2008 to 2012, and the blood was stored at -81°C after centrifugation. Due to the insufficient amount of material collected, we re-collected material for the study in 2019 to 2023. The obtained plasma was also stored at -81°C until assays. In both rounds, the inclusion criteria were the same
-What were the exclusion criteria for the study group?
Thank You for Your comment.
The primary exclusion factor from the study was a diagnosis of cancer other than ovarian cancer. We also did not include patients who had previously received chemotherapy or radiotherapy, or patients with renal failure (this disease is associated with impaired HE4 clearance resulting in high levels of this marker). Exclusion criteria are shown in the newly added Figure 9.
-It is not clear when the blood for biochemical analysis was drawn? Was it before or after diagnosis?
Thank You for Your comment.
Blood for biochemical analysis was taken immediately after the patients were included in the study, in the case of women with ovarian cancer before the inclusion of treatment - in such a way that chemotherapy did not affect the concentration of the markers tested. Biochemical marker determinations were performed when we received the histopathological diagnosis. This is shown in detail in Figure 9.
Again, we thank the reviewer for all the guidance and corrections. It is our hope that the manuscript, after revision, will meet the reviewer's expectations and be published in the International Journal of Molecular Sciences.
Best regards,
prof. dr hab. Sławomir Ławicki
also, on behalf of all authors

Reviewer 4 Report
Comments and Suggestions for Authors
The work presented is highly interesting and significant for scientific purposes. In fact, the poor sensitivity of the tests and less studied features of ovarian cancer devoid a prevention, diagnosis and prognosis, as well as the effectiveness of the therapies or the choise of an useful therapeutic window.
The authors investigate an innovative and very useful correlation, which enriches the panel of knowledge about this form of cancer. Given the small number of such evidence, I think that any information of this kind is very important.
The work is presented in a very clear way, with a smooth and simple writing style, despite the complexity of the analysis of correlations conducted.
comments:
The introduction would benefit if it were enriched with a brief discussion on the current therapeutic strategies against ovarian cancer, and with what effectiveness.
likewise I recommend that authors include in this section if there are clinical trials or preclinical in vivo/vitro studies that have evaluated MMPs, finding similar evidence. This in the discussion is very well explained, but I believe that in the speech on MMPs would benefit from information about his analysis in other clinical trials
line 77-86 and 87-96 I think it’s a repeat caused by a typo
Table 1: In the captions of the table or graphs it should be inserted that the concentration data are derived from ELISA or other tests.
The correlations carried out are very interesting and well developed, on a large cohort.
Materials and methods: perhaps a flowchart of the approaches applied in the work would help the reader to follow schematically the steps and the correlations carried out by the authors
Figure 7: In the OC group there are many outliers, how would the authors explain and contextualize this modulation?
Author Response
Reviewer 4.
The work presented is highly interesting and significant for scientific purposes. In fact, the poor sensitivity of the tests and less studied features of ovarian cancer devoid a prevention, diagnosis and prognosis, as well as the effectiveness of the therapies or the choise of an useful therapeutic window.
The authors investigate an innovative and very useful correlation, which enriches the panel of knowledge about this form of cancer. Given the small number of such evidence, I think that any information of this kind is very important.
The work is presented in a very clear way, with a smooth and simple writing style, despite the complexity of the analysis of correlations conducted.
Dear reviewer.
We sincerely thank You for such a positive review of our paper - „Diagnostic utility of selected matrix metalloproteinases (MMP-2, MMP-3, MMP-11, MMP-26), HE4, CA125 and ROMA algorithm in diagnosis of ovarian cancer”. We will try our best to answer each reviewer's question accurately and honestly. Answers will be written in green italics, while we will mark changes in the manuscript in light green.
comments:
-The introduction would benefit if it were enriched with a brief discussion on the current therapeutic strategies against ovarian cancer, and with what effectiveness.
Thank You for Your comment.
We agree that discussion of therapeutic strategies in ovarian cancer will enrich our manuscript - a relevant paragraph has been added to the paper (lines 71-79). Treatment of ovarian cancer is mainly based on surgical methods and chemotherapy. The information for this paragraph is partly derived from works already cited:
-World Ovarian Cancer Coalition. World Ovarian Cancer Coalition Atlas 2023. Available at: https://worldovariancancercoalition.org/wp-content/uploads/2023/03/World-Ovarian-Cancer-Coalition-Atlas2023-FINAL.pdf. Accessed April 2024.
- Ravindran, F.; Choudhary, B. Ovarian cancer: molecular classification and targeted therapy. IntechOpen, 2021, 1–21.
We also added two new works for the new section:
- Trimbos, J.B. Surgical Treatment of Early-Stage Ovarian Cancer. Best Practice & Research Clinical Obstetrics & Gynaecology 2017, 41, 60–70, doi:10.1016/j.bpobgyn.2016.10.001.
- González-Martín, A.; Harter, P.; Leary, A.; Lorusso, D.; Miller, R.E.; Pothuri, B.; Ray-Coquard, I.; Tan, D.S.P.; Bellet, E.; Oaknin, A.; et al. Newly Diagnosed and Relapsed Epithelial Ovarian Cancer: ESMO Clinical Practice Guideline for Diagnosis, Treatment and Follow-Up. Annals of Oncology 2023, 34, 833–848, doi:10.1016/j.annonc.2023.07.011.
-likewise I recommend that authors include in this section if there are clinical trials or preclinical in vivo/vitro studies that have evaluated MMPs, finding similar evidence. This in the discussion is very well explained, but I believe that in the speech on MMPs would benefit from information about his analysis in other clinical trials
Again, thank You for Your comment.
It is unfortunate that at this point we do not have preclinical or clinical studies on MMPs. It is worth noting, however, that some of the preclinical studies have been on MMPs inhibitors with anti-angiogenic properties. The first synthetic inhibitor of MMPs was Batimastat. However, this compound did not pass Phase III clinical trials (doi: 10.1002/stem.170237). Another compound was Marimastat, however, it achieved poor results in clinical trials (doi: 10.1111/j.1749-6632.1999.tb07688.x). Although synthetic inhibitors of MMPs have achieved good results in animal models, they are not currently used in human cancer therapy. Aside from a number of side effects, these compounds had no effect on tumor progression (doi: 10.1158/1535-7163.MCT-17-0646).
As for in vitro studies, there are few papers identifying the role of individual MMPs in ovarian cancer. Most of the studies define the role of MicroRNA silencing of individual MMPs However, we will not include a description of in vitro activities, as in our opinion they will not substantively enrich our manuscript, since we do not refer to in vitro studies.
-line 77-86 and 87-96 I think it’s a repeat caused by a typo
Thank You for Your comment and we sincerely apologize for the mistake. We unintentionally inserted the same paragraph twice in the manuscript. We have removed the mistake.
-Table 1: In the captions of the table or graphs it should be inserted that the concentration data are derived from ELISA or other tests.
Thank You for Your comment.
Both the relevant tables and figures have been enriched with information about the origin of the concentrations in question. All changes to the manuscript have been highlighted in green.
-The correlations carried out are very interesting and well developed, on a large cohort.
Thank You very much for such a positive comment.
-Materials and methods: perhaps a flowchart of the approaches applied in the work would help the reader to follow schematically the steps and the correlations carried out by the authors
Thank You for Your comment. As rightly suggested by the reviewer, a flow chart has been added to the manuscript. It is labeled as Figure 9 and quoted in the text.
-Figure 7: In the OC group there are many outliers, how would the authors explain and contextualize this modulation?
Thank You for Your comment.
We realize that there are some outliers in the OC patient group. In accordance with the ethics of scientific research, we did not remove the outliers. Outliers indicate a higher concentration of MMPs in a particular patient's blood. We cannot say with certainty what these variations are due to, however, we postulate that it may be due to, for example, the presence of potential undetected comorbidities (e.g., hypertension), stimulants, and personal conditions. However, we can certainly exclude the influence of chemotherapy because blood samples were taken before the introduction of any treatment, and the influence of kidney disease - because of the significant effect of kidney disease on HE4 values, patients with kidney failure were not eligible for the study. However, unequivocally determining the role of the described factors on MMPs concentrations requires further research.
Again, we thank the reviewer for all the guidance and corrections. It is our hope that the manuscript, after revision, will meet the reviewer's expectations and be published in the International Journal of Molecular Sciences.
Best regards,
prof. dr hab. Sławomir Ławicki
also, on behalf of all authors

Round 2
Reviewer 1 Report
Comments and Suggestions for Authors
All comments have been addressed.
Author Response
Dear Reviewer,
Thank You again for reviewing and positively evaluating our revisions.
Best regards,
prof. dr hab. Sławomir Ławicki
also, on behalf of all authors
Reviewer 3 Report
Comments and Suggestions for Authors
No further comments
Author Response

(The authors gave the same response as above.)
